# Plant Immunity against Tobamoviruses

**DOI:** 10.3390/v16040530

**Published:** 2024-03-29

**Authors:** Xiyin Zheng, Yiqing Li, Yule Liu

**Affiliations:** 1MOE Key Laboratory of Bioinformatics and Center for Plant Biology, School of Life Sciences, Tsinghua University, Beijing 100084, China; 2Tsinghua-Peking Center for Life Sciences, Beijing 100084, China

**Keywords:** plant immunity, tobamovirus, RNA silencing, RNA decay, NLR, phytohormone, genome editing

## Abstract

Tobamoviruses are a group of plant viruses that pose a significant threat to agricultural crops worldwide. In this review, we focus on plant immunity against tobamoviruses, including pattern-triggered immunity (PTI), effector-triggered immunity (ETI), the RNA-targeting pathway, phytohormones, reactive oxygen species (ROS), and autophagy. Further, we highlight the genetic resources for resistance against tobamoviruses in plant breeding and discuss future directions on plant protection against tobamoviruses.

## 1. Introduction

Tobamoviruses are a group of plant viruses belonging to the *Tobamovirus* genus, which is part of the *Virgaviridae* family. The *Tobamovirus* genus includes 37 members [1]. The tobamovirus genome is a single-stranded, positive-sense RNA of about 6.4 kb size that is encapsidated in rod-shaped particles. It encodes four viral proteins, including two subunits of the viral RNA-dependent RNA polymerase-a 5′, where there is a proximal one of 122–130 kDa and a translational read through of 178–183 kDa from the first open reading frame (ORF); a movement protein (MP); and a coat protein (CP) [2]. In addition, tobacco mosaic virus (TMV) contains two additional ORFs which potentially encode a 54 kDa and 4.8 kDa protein, respectively [3,4]. Tobamoviruses infect a wide range of plant species and cause significant damage to a wide range of economically important crops, such as tomato, pepper, cucumber, and tobacco.

In this review, we discuss the plant immunity responses to tobamoviruses, including pattern-triggered immunity (PTI), effector-triggered immunity (ETI), RNA-targeting pathway, phytohormones, reactive oxygen species (ROS), and autophagy. Further, we highlight the genetic resources for resistance against tobamoviruses in plant breeding and provide future directions on plant protection against tobamoviruses.

## 2. Plant Defense against Tobamoviruses

Based on the studies on plant interactions with bacterial and fungal pathogens, plants are thought to employ a two–layer immune system consisting of PTI and ETI. PTI is triggered by pathogen-associated molecular patterns (PAMPs) via cell surface-localized pattern-recognition receptors (PRRs), whereas ETI is activated by pathogen effector proteins via predominantly intracellularly localized receptors called nucleotide-binding, leucine-rich repeat immune receptors (NLRs) [5]. The concept of PTI and ETI could also be applicable to plant-virus interactions.

PTI is triggered by PRRs to detect the conserved microbial cues known as PAMPs. Classic PRRs are localized to the plasma membrane (PM) and play a crucial role in plant immunity against extracellular fungi and bacterial pathogens [6]. As viruses are intracellular parasites, it seems unlikely that there is a classic cell-surface-localized PRR-mediated PTI against intracellular viruses. However, antiviral PTI has been proposed because some viral proteins suppress PTI responses activated by non-viral PAMPs, and some receptor-like kinases are involved in basal antiviral defense [7]. For example, Brassinosteroid-associated kinase 1 (BAK1), the core regulator of PTI, contributes to plant defense against several RNA viruses, including two tobamoviruses, TMV and oilseed rape mosaic virus [8]. Viral double-stranded RNAs (dsRNAs) have been proposed to be the potential PAMPs that induce the antiviral defense response. Indeed, plant treatment with the purified viral dsRNA from virus-infected plants and the dsRNA analog polyinosinic:polycytidylic acid triggers typical PTI responses and antiviral defense, which depend on the PTI co-receptor SOMATIC EMBRYOGENESIS RECEPTOR-LIKE KINASE 1 [8]. Small RNAs (sRNAs) stimulate the production of callose near plasmodesmata (PD), consequently restricting the spread of viruses between plant cells. This defense response relies on various PTI signaling elements and several PD-localized proteins [9]. The expression of TMV MP suppresses dsRNA-induced callose deposition, facilitating intercellular TMV movement [10]. In addition, Ca^2+^ flux triggered by injuries to plant cells is thought to be the common molecular pattern of different viral infections which primes antiviral RNAi defense [11]. These findings suggest the existence of non-classic rather than classic PTI against plant viruses, including tobamoviruses.

Plant NLRs play an important role in plant antiviral ETI by detecting viral effectors. These NLRs can recognize viral effectors and trigger antiviral ETI, usually including hypersensitive response (HR), a type of programmed cell death at pathogen infection sites [12]. Currently, most plant genetic resources for plant breeding are mainly controlled by the naturally identified dominant resistance genes including *NLR* genes, *Tm-1*, and recessive resistance genes (see Figure 1 and Table 1).

## 3. Plant Genetic Resources for Resistance against Tobamoviruses

### 3.1. The Tobacco N Gene

The *N* gene is the first *resistance (R)* gene identified in *Nicotiana glutinosa* [13,14]. The *N* gene confers resistance to all known tobamoviruses except TMV-Ob [15,16]. It is a toll-interleukin-1 receptor homology/nucleotide binding/leucine rich repeat (TIR-NB-LRR, TNL) class of resistance gene [17]. The structure-function analysis shows that all three domains of the N protein, including TIR, NB, and LRR, are necessary for its function [18]. The *N* gene encodes two transcripts, N_S_ and N_L_, by alternative splicing, both of which are essential for the full resistance to TMV [19]. TMV induces HR lesions in *N*-containing plants. In *N*-containing plants, the burst of reactive oxygen intermediates occurs rapidly upon TMV infection [20]. Similarly, nitric oxide accumulates at the beginning of HR during TMV infection [21].

The N protein recognizes the 50 kDa helicase domain (p50) within the 126 kDa replicase subunit of tobamoviruses through its TIR domain, triggering the HR and immune response in the cytoplasm [22,23,24,25]. Simultaneously, the N protein also functions in the resistance to TMV in the cell nucleus [23,26]. Extensive research has revealed the roles of host regulators in *N*-mediated resistance primarily through their interaction with the N protein. The transcription factor SQUAMOSA PROMOTER BINDING PROTEIN-LIKE 6 (SPL6) associates with the N protein within distinct nuclear compartments, and it is essential for *N*-mediated resistance against TMV. In particular, the N-SPL6 interaction is present in the nucleus and is detected only when the p50 is present. It suggests that the association of N with SPL6 only occurs after an active defense response [26]. Similar to other TNLs, the N function is ENHANCED DISEASE SUSCEPTIBILITY (EDS1)-dependent [27,28]. N requirement gene 1 (NRG1) serves as a helper NLR and is required for *N*-mediated resistance [29]. Rar1 is required for N function. The tobacco Rar1 interacts with SGT1, a novel subunit of the SCF-type (Skp1/Cullin/F box protein) E3 ubiquitin ligase complex involved in protein degradation. SGT1 and Rar1 associate with Heat shock protein 90 (Hsp90), which interacts with the N protein [30]. Moreover, Hsp90 suppression compromises *N*-mediated resistance to TMV [31]. Similarly, the Hsp40-like Dna-J domain movement protein-interacting proteins (MIP1s) interact with SGT1 and are required for *N*-mediated resistance [32]. N receptor-interacting protein (NRIP1), a functional rhodanese sulfurtransferase, has been identified to directly interact with both the N TIR domain and TMV p50, which is necessary for a complete resistance to TMV. NRIP1, which is normally localized in chloroplasts, is recruited to the cytoplasm and nucleus by the p50 effector. Consequently, NRIP1 interacts with N only in the presence of p50 [33]. UBR7, a putative E3 ubiquitin ligase, directly interacts with the N protein via its TIR domain, and it also negatively regulates the level of the N protein. The downregulation of UBR7 increases the protein level of N and enhances TMV resistance. Moreover, TMV p50 disrupts the N-UBR7 interaction and relieves the negative regulation on N [34]. In addition, the MEKK1-like mitogen-activated protein kinase kinase kinase NPK1, MEK1 MAPKK, NTF6 MAPK, and WRKY/MYB transcription factors are essential for *N*-mediated resistance [35,36]. Transcription factor alfin-like 7 (AL7) interacts with N and inhibits the transcription of genes involved in ROS scavenging to positively regulate *N*-mediated resistance to TMV [37]. Mitogen-activated protein kinases (MAPKs), salicylic acid-induced protein kinase (SIPK), and wound-induced protein kinase (WIPK) interact with and phosphorylate AL7, which inhibits AL7-N interaction and enhances its DNA binding activity, thus promoting ROS accumulation and enabling an immune response to TMV [37].

### 3.2. The Tobacco N′ Gene

The *N′* gene from *N. sylvestris* encodes a coiled-coil (CC) domain-containing NLR (CNL) immune receptor that confers resistance against tobamoviruses accompanying the HR by recognizing viral CP [38,39,40,41]. *N′* is an ortholog of the pepper *L* genes with a different recognition spectrum. *N′* can confer resistance against tomato mosaic virus (ToMV), paprika mild mottle virus, pepper mild mottle virus (PMMoV), including the PMMoV pathotype P_1,2,3,4_ [41] and tomato brown rugose fruit virus (ToBRFV) [16].

### 3.3. The Tomato Tm-1 Gene

The *Tm-1* gene is introgressed into the cultivated tomato species *Solanum lycopersicum* from the wild tomato species *Solanum habrochaites* S. Knapp & D.M. Spooner [42,43]. However, *Tm-1*-mediated ToMV resistance could be easily overcome by ToMV resistance breaking isolates [44]. In addition, *Tm-1* suppresses the visible viral mosaic symptoms, but detectable virus multiplication still occurs. The inhibition of TMV multiplication is *Tm-1* gene dose-dependent, whereas the suppression of visible symptoms is not [45,46].

The *Tm-1* gene encodes a protein that binds to the 130 kDa subunit of the ToMV replicase and inhibits the RNA-dependent RNA replication of ToMV [47,48], suggesting that Tm-1 functions as a viral inhibitor. Moreover, structure analyses have indicated that Tm-1 shows no sequence homology to functionally characterized proteins [47,49]. These results indicate that *Tm-1* differs from the previously identified resistance genes in plants.

Tm-1 binds ToMV replication proteins to inhibit the key events in replication complex formation on membranes, preceding negative-strand RNA synthesis. Three host proteins, Tobamovirus multiplication 1 (TOM1), TOM2A, and ADP-ribosylation factor-like 8 (ARL8), are required for ToMV RNA replication, and they are also suggested to be the components of the ToMV replication complex [50]. Upon ToMV infection, Tm-1 inhibits the formation of viral RNA replication complex on membranes by inhibiting the association of TOM1, TOM2A, and ARL8 with the ToMV 130 K replicase component [48].

### 3.4. The Tomato Tm-2/Tm-2^2^ Gene

*Tm-2* and *Tm-2^2^* are two alleles of the same gene in tomato that encodes a CNL [51]. They have been introgressed into cultivated tomato from the wild species *Solanum peruvianum. Tm-2* and *Tm-2^2^* confer resistance to tobamoviruses, including TMV and ToMV, by recognizing the viral MP [42,52,53,54]. *Tm-2^2^* also displays resistance to tomato mottle mosaic virus (ToMMV) by recognizing the MP, and the resistance is regulated by the allele combinations and the temperature [55]. The homozygous tomato harboring *Tm-2^2^* and the heterozygous tomato containing *Tm-2^2^* and *Tm-2*, but not the heterozygous tomato containing *Tm-2^2^* and *Tm-2*, exhibit resistance to ToMMV. *Tm-2^2^*-mediated resistance is compromised at 35 °C, but not at 30 °C or lower temperatures [55]. Previous studies showed that the C-terminal 30-amino acid deletion of viral MP compromised *Tm-2^2^*-mediated resistance [54]. However, we showed that the N-terminus of MP is sufficient for inducing *Tm-2^2^*-mediated HR [56]. Further, ToBRFV can overcome *Tm-2^2^*–mediated resistance, and the elements required to evade Tm-2^2^ are located in the N-terminus but not in the C terminus of ToBRFV MP, as proved by virus infection assays [57,58]. These combined data suggest that Tm-2^2^ recognizes the N-terminal but not the C-terminal sequence of the viral MP for *Tm-2^2^* recognition, and the C-terminal domain of MP could affect the exposure of protein structures that are recognized by Tm-2^2^. Tm-2^2^ is previously thought to function on PD due to the predominant localization of its avirulence (Avr) protein MP. However, the Tm-2^2^ function is independent of the localization of viral MP to PD. Further, Tm-2^2^ has been found to localize to and function on the PM although it lacks any PM-localization motif [56]. The Tm-2^2^ CC domain is the signaling domain and its self-association triggers a defense response, including HR. In the presence of viral MPs, Tm-2^2^ self-associates and is activated, which requires the nucleotide-binding domain-mediated self-association of a CC domain in an (d)ATP-dependent manner [59]. Tm-2^2^ stability is regulated by SGT1 and Hsp90 [60] and MIP1s [32]. MIP1s function as co-chaperones and are required for both TMV infection and plant immunity, including *Tm-2^2^*-mediated resistance, by associating with SGT1 and Tm-2^2^ [32]. Further, rubisco small subunit is required for both *Tm-2^2^*-mediated extreme resistance and tobamovirus movement by interacting with viral MPs [61]. In addition, the *Tm-2^2^*-mediated resistance response is dependent on its expression level: a high level of expression triggers extreme resistance without visible cell death; an intermediate level of expression triggers complete resistance with HR lesions at virus infection sites; and a low level of expression only confers a partial resistance with systemic viral infection and systemic necrosis throughout the plant [62].

### 3.5. The Pepper L Gene

The *L* gene confers resistance against tobamovirus and encodes a CNL, which are introgressed into *Capsicum annuum* from the wild pepper species [63]. Upon virus infection, the L protein recognizes viral CP, leading to HR [64]. There are four different alleles of *L* (*L^1^*, *L^2^*, *L^3^*, and *L^4^*). All *L* alleles confer resistance to P_0_ viruses, including ToMV, yellow pepper mild mottle virus, and chili pepper mild mottle virus. *L^1^* only confers resistance to P_0_ viruses. *L^2^* confers resistance to all P_0_ and P_1_ viruses, including the PMMoV J strain. *L^3^* defends against P_0_, P_1_, and P_1,2_ viruses, including the PMMoV strains which can overcome *L^2^*. *L^4^* defends against P_0_, P_1_, P_1,2_, and P_1,2,3_ PMMoV pathotypes which can overcome *L^3^* [63,64,65]. However, some PMMoV strains can systemically infect all the identified *L* alleles of pepper plants. Another allele, *L^1a^*, is thermosensitive and does not confer resistance against tobamoviruses at elevated temperatures [66].

### 3.6. Tobamovirus Multiplication (TOM) and ARL8

Host susceptibility proteins help virus infection at different stages of the virus life cycle. Among them, TOBAMOVIRUS MULTIPLICATION (TOM) proteins play a critical role in tobamovirus infection by interacting with viral replication-associated proteins to help the formation of the viral replication complex [67]. *TOM* genes are critical for infection by tobamoviruses in various plant species [68]. TOM1 and TOM2A encode seven-pass and four-pass transmembrane proteins, respectively [69,70]. *TOM1* was first cloned from *Arabidopsis* with three homologues, *TOM1*, *TOM3*, and *THH1* [70,71]. In *Arabidopsis*, the *tom1* single mutant partially impairs tobamovirus multiplication, and the double *tom1* and *tom3* mutant completely inhibits tobamovirus multiplication [72,73]. In double *tom1/tom3* mutant lines overexpressing *THH1*, the level of tobamovirus CP is similar to that of wild-type plants, suggesting that THH1 could weakly contribute to tobamovirus multiplication due to its lower level of expression than that of TOM1 and TOM3 [71].

*TOM1* homologs have also been found in tobacco and tomato [74,75]. Knockdown of *TOM1* homologs dramatically inhibits TMV/ToMV multiplication without introducing any obvious growth defects [74,75,76]. In addition, tobacco *TOM1* mutant lines have been found to be resistant to TMV [77]. In particular, the quadruple knockout of *SlTOM1* homologs, which is generated by genome editing, confers resistance to ToBRFV in tomato [78]. Meanwhile, the double knockout of *SlTOM1a* and *SlTOM3* confers resistance to ToBRFV, but not to ToMV and TMV [67]. 

TOM1 and TOM2A promote tobamovirus multiplication. Upon TMV infection, TOM1 interacts with TMV 126-kDa replicase to promote the assembly of viral replication complex formation on host membranes. TOM2A also facilitates the formation of the viral replication complex by interacting with TOM1 [68].

Arabidopsis ADP-ribosylation factor-like 8 (ARL8) is a small GTP-binding protein that interacts with TOM1. ARL8 also interacts with ToMV 180-kDa replicase and is required for tobamovirus multiplication. Upon tobamovirus infection, ARL8 and TOM1 are components of the replication complex and play crucial roles in the replication activation process, including replicase RNA synthesizing and capping [50,79].

### 3.7. WPRb

*WPRb*, a weak chloroplast movement under blue light 1 and plastid movement impaired 2 (WEB1/PMI2)-related protein family gene, is a recessive resistance gene associated with cucumber green mottle mosaic virus (CGMMV) resistance in watermelon. Genome editing of *WPRb* in *N. benthamiana* also confers a great tolerance to CGMMV. WPRb targets the PD and interacts with CGMMV MP to facilitate viral cell-to-cell movement by affecting PD permeability [80].

**Table 1 viruses-16-00530-t001:** Plant genetic resources for resistance against tobamoviruses.

Gene Name	Alleles	Plant of Origin	Viral Target	Protein Type
*N*	NA	*Nicotiana glutinosa* [13,14]	p50 (Avr) [24,25,26]	Toll-interleukin-1 receptor homology/nucleotide binding/leucine rich repeat (TNL) [17]
*N′*	NA	*Nicotiana sylvestris* [41]	CP (Avr) [41]	coiled-coil domain-containing/nucleotide binding/leucine rich repeat (CNL) [41]
*Tm-2*	*Tm-2* and *Tm-2^2^* [51]	*Solanum peruvianum* [42,52]	MP (Avr) [54,56]	CNL [51]
*L*	*L^1^*, *L^1^^a^*, *L^2^*, *L ^3^*, and *L^4^* [64,66]	*Capsicum chinense* [63]	CP (Avr) [64]	CNL [64]
*Tm-1*	Several [49]	*Solanum habrochaites* S. Knapp & D.M. Spooner [42,43]	Replicase [44,46,49]	Unidentified
*TOM1*; *TOM3*	NA	*Arabidopsis* [72,73]	Replicase [50,68,79]	Seven-pass transmembrane [70]
*TOM2A*	NA	*Arabidopsis* [69]	NA	Four-pass transmembrane [69]
*ARL8*	NA	*Arabidopsis* [79]	Replicase [50,68,79]	N-terminal amphipathic helix [79]
*WPRb*	NA	*Citrullus lanatus* [80]	MP [80]	coiled-coil [80]

Abbreviations: NA, not available.

## 4. RNA-Targeting Mechanisms

RNA silencing plays a key role in antiviral defense against all types of viruses [81,82]. RNA silencing is a sequence-specific process found in both plants and animals, which involves the generation of sRNAs [83,84]. In plants, RNA silencing is orchestrated by 21- to 24-nucleotide sRNA, which is categorized as small interfering RNAs (siRNAs) and microRNAs (miRNAs). These sRNAs are generated as duplexes with 2-nt 3′ overhangs from longer dsRNA precursors or hairpin-like secondary structures, respectively, through the action of Dicer-like (DCL) enzymes [85] (Figure 2). RNA silencing has been used to engineer complete resistance against tobamoviruses [86,87,88,89,90,91]. In the case of tobamoviruses, the small replicase subunit (122–130 kDa) could function as the viral suppressors of RNA silencing (VSR). In particular, the TMV 126 kDa protein is identified as the VSR by disrupting HUA enhancer 1 (HEN1)-mediated methylation of sRNAs to shield viral transcripts from the host RNA silencing pathway [92,93]. In addition, the 122-kDa replicase subunit (p122) of crucifer-infecting TMV (crTMV) is a potent VSR and compromises both siRNA- and miRNA-mediated pathways [94]. p122 is also reported to enhance the levels of *miRNA 168* to inhibit the expression of Argonaute 1 (AGO1) [95]. In addition, during oilseed rape mosaic tobamovirus (ORMV) infection, ORMV p125 replicase is required for the inhibition of HEN1 activity to suppress RNA silencing [85]. On the other hand, TMV MP contributes to antiviral silencing during infection by enhancing the spread of RNA silencing signal, and this ability of TMV MP may contribute to the control of virus propagation in the infected host. The TMV 126 kDa replicase-associated protein and MP with contrast roles in RNA silencing may balance viral propagation at different infection stages [96].

Beyond RNA silencing, other RNA-targeting mechanisms have also been shown to be involved in antiviral defense including RNA decay [97]. RNA decay is an essential RNA quality control and gene regulatory mechanism in eukaryotes. It is initiated in the cytoplasm by mRNA deadenylation, followed by exosome complex-mediated exonucleolytic decay in the 3′-5′ direction or by decapping complex and exoribonuclease (XRN)-mediated decay in the 5′-3′direction [98]. A study suggests that TMV proteins (MP and CP) enhance transcriptional levels of RNA decay genes and induce RNA decay to impair antiviral RNA silencing for better virus infection [99]. However, silencing of *NbXrn4* facilitates TMV systemic infection in *N. benthamiana* [100], suggesting that RNA decay may also play a role in antiviral defense against tobamovirus.

## 5. Phytohormone Interactions with Tobamoviruses

Plant hormone salicylic acid (SA), methyl salicylate (MeSA), jasmonic acid (JA), methyl jasmonate (MeJA), ethylene, and Auxin/indole-3 acetic acid (Aux/IAA) play important roles in plant–virus interactions. Among them, SA plays a critical role in plant defense against a broad spectrum of pathogens, including multiple viruses. SA interferes with different steps of the viral cycle. In tobacco leaves, SA treatment decreases TMV RNA accumulation by disrupting TMV replication in mesophyll cells [101,102]. SA also inhibits TMV cell-to-cell movement [102]. Reducing the early SA accumulation delays HR and promotes TMV dispersal during lesion formation in TMV-infected tobacco Samsun *NN*, suggesting that early SA accumulation is a key factor in preventing viral escape during *N*-mediated TMV resistance [103]. In addition, SA could also function its antiviral mechanism by activating RNA silencing. SA induces the expression of *host RNA-dependent RNA polymerase1* (*RDR1*), which contributes to antiviral RNA silencing, thereby promotes the degradation of viral RNA to limit the infection by tobamovirus [104,105,106]. Further, SA upregulates the expression of *RDR6* in *N. tabacum* [107] and *RDR2* in tomato [108]. However, it does not affect the expression of *RDR2* and *RDR6* in *Arabidopsis* [109], nor does it affect *RDR6* in *N. glutinosa* [110]. Besides SA, JA/MeJA, ABA, ET, and synthetic auxin upregulate *RDR1* expression [111,112]. ABA treatment also increases the expression of *RDR1*, *RDR2*, and *RDR6* when SA synthesis is impaired, suggesting that the antagonism between ABA and SA influences the expression of these genes [112]. *N*-mediated TMV resistance is compromised in transgenic *NahG* tobacco plants and *NPR1*-silenced plants [35,113], suggesting that SA is essential for *N*-mediated antiviral immunity.

TMV infection induces HR accompanied by the production of phytohormones, including SA and JA. Exogenous MeJA application to plants reduces local TMV resistance and permits systemic TMV movement [114], suggesting a negative role of MeJA in antiviral defense against TMV. However, silencing of JA biosynthetic and signaling genes in *N. benthamiana* plants increases susceptibility to TMV [115], suggesting a positive role of MeJA/JA in antiviral defense against TMV. The role of COI1 in *N*-mediated TMV resistance is also controversial. Silencing of *COI1*, the JA receptor gene, compromises *N*-mediated TMV resistance in transgenic *N. benthamiana* [35], suggesting that JA positively regulates *N*-mediated TMV resistance. However, silencing of *COI1* reduces viral accumulation in *N. tabacum* Samsun *NN*, which possesses the *N* gene [114]. The contrast data may be a result of the differences in the experimental plant systems used, i.e., *N. tabacum* Samsun *NN* and *N. benthamiana*. Indeed, *N. tabacum* Samsun *NN* is reported to have a novel *N* gene-associated, temperature-independent resistance [116].

TMV infection triggers the plant release of several airborne compounds including (E)-2-octenal. (E)-2-octenal primes the JA/ET signaling pathway, including the upregulation of *NbMYC2*, *NbERF1*, and *NbPDF1.2*, and then upregulates the pathogenesis-related genes, such as *NbPR1a*, *NbPR1b*, *NbPR2*, and *NbNPR1*, to activate the antiviral defense against TMV in adjacent *N. benthamiana* plants [117].

Both MeSA and MeJA contribute to systemic resistance against TMV, possibly acting as the initiating signals for systemic resistance. Silencing of SA or JA biosynthetic and signaling genes in *N. benthamiana* plants increases susceptibility to TMV [115]. Silencing of either *SABP2* or *NAC2* compromises antiviral defense, suggesting that SA, but not MeSA, directly activates antiviral defense [118].

Ethylene plays a role in the antiviral defense against tobamoviruses. It increases watermelon resistance to CGMMV infection by inducing the expression of the *AGO5* gene [119]. In addition, the ethylene pathway participates in transcription factor MYB4L-mediated resistance against TMV, and ethylene-induced MYB4L is involved in the TMV resistance in *N. benthamiana* [120]. Moreover, silencing of *CTR1*, an ethylene receptor, accelerates *N*-mediated HR [35], suggesting that ethylene signaling negatively regulates the *N*-mediated HR induced by TMV.

Auxin is a crucial plant hormone and participates in various processes. Aux/IAA proteins are vital components within this regulatory framework, with a primary function of translating auxin levels into gene expression [121]. The interaction between tobamoviruses and Aux/IAA was first reported to involve the helicase domain of TMV replicase and IAA26. The expression of TMV replicase disrupts the nuclear localization of IAA26, inhibiting its putative function as a transcriptional regulator of auxin-responsive genes for better viral symptoms and systemic movement [122,123,124]. TMV can reprogram auxin/IAA protein transcriptional responses and then enhances virus phloem loading [125].

## 6. ROS in Tobamovirus Infection

Upon pathogen infection, plants rapidly produce ROS to induce local or systemic signaling through the activation of cell surface-localized respiratory burst oxidase homolog (RBOH) proteins. ROS signaling mediates systemic resistance against plant viruses and is often considered as a positive regulator of plant antiviral defense [126,127]. Intact TMV virion and isolated TMV CPs trigger a rapid oxidative burst when added to the apoplast of tobacco epidermal cells. TMV CP stimulates host NAD(P)H oxidase-like activity [128]. Meanwhile, TMV infection increases the expression of ROS-scavenging related genes, including superoxide dismutases (CSD2), ascorbate peroxidase (APX1), and GDP-mannose pyrophosphorylase 1 (GMP1). Furthermore, silencing of *GMP1* enhances the ROS level and reduces the TMV accumulation [129,130]. Thus, the ROS-scavenging pathway can also modulate the plant resistance against tobamoviruses. In addition, type-I non-specific lipid transfer protein (LTP1) is reported to enhance antiviral defense against TMV by upregulating SA biosynthesis and its downstream signaling components, while also suppressing ROS accumulation during the later stages of viral pathogenesis [131].

## 7. Autophagy in Tobamovirus Infection

Autophagy is a conserved vacuole/lysosome-dependent cellular process mediating the degradation of senescent/dysfunctional organelles or cytoplasmic materials into peptides or amino acids for reuse or storage [132]. Autophagy functions as an antiviral mechanism in plants [133,134]. It may also limit tobamovirus infection. Autophagy is activated upon infection by PMMoV or TMV [135,136]. PMMoV infection upregulates the expression of multiple autophagy-related genes (ATGs). Disruption of autophagy by autophagy inhibitor treatment and silencing of *ATG* genes increases PMMoV RNA accumulation and aggravates viral systemic symptoms [135]. Although increased autophagy by silencing of *Cytoplastic Glyceraldehyde-3-Phosphate Dehydrogenase* genes has no effect on GFP-tagged TMV infection, it promotes *N*-mediated HR cell death [137]. Further, autophagy suppression negatively regulates *N*-mediated cell death and promotes local TMV infection in *N*-containing plants [138]. Similarly, Bax inhibitor-1 positively regulates the autophagy triggerd by *N* activation upon TMV infection and negatively regulates *N*-mediated cell death [139]. In addition, autophagy also regulates the ROS level and PCD progress in TMV-infected tomato plants [140].

## 8. Conclusions

There are several antiviral defense mechanisms against tobamovirus infection, including the non-classic PTI, ETI, RNA-targeting pathway, phytohormones, ROS, and autophagy. Understanding how they function and interact will help to control plant viral diseases. Additionally, some recessive resistance genes also play a valuable role in this regard.

## 9. Future Directions

In the last few decades, a number of studies have demonstrated that transgenic approaches, especially by enhancing RNA silencing against viral RNA sequences, can provide effective plant protection against viruses [141]. Due to public concerns and strict regulatory barriers, this approach has been restricted for field use. However, it still holds great potential for generating virus-resistant plants.

Genome editing is another breakthrough for generating efficient virus resistance. Editing of tobamovirus susceptibility genes based on CRISPR/Cas9 systems can be used to generate crops with resistance against tobamoviruses. Numerous host susceptibility genes such as *TOM1/3*, *TOM2*, *ARL8*, and *WPRb*, have been identified, and their knockouts or mutants do not have any obvious effect on plant growth or morphology. More recently, Kunitz peptidase inhibitor-like protein (KPILP) was identified as a novel proviral factor during TMV or crTMV infection [142]. Editing of these genes may achieve tobamovirus resistance in different crops [143].

Engineering NLRs with new recognition targets is another approach that can be applied for protection against viruses [143]. The engineered NLRs can be generated for recognizing new pathogens by several technologies, including protein engineering, random or site-directed mutagenesis, and structure-based predictions. These strategies may be used to develop new crop resistance against tobamoviruses.

The exogenous application of dsRNA to induce RNA silencing has been perceived as an alternative to transgenesis and can be used for plant protection against viral disease. It has been shown to provide some protection against TMV [144] and can be modified to fight other tobamoviruses. However, the efficiency of this approach is affected by several factors, including the concentration/dose and length/size of dsRNAs, application method, delivery technique, plant organ-specific activities, and stability under open-field conditions [145]. Technical advances in these fields will overcome these restrictions for agricultural application. In addition, cross-protection is another efficient strategy without genetic modification, which is able to fight against severe virus strains. Reverse genetics can be adopted to generate attenuated mutants that have potential in cross-protection against tobamoviruses [141].

## Figures and Tables

**Figure 1 viruses-16-00530-f001:**
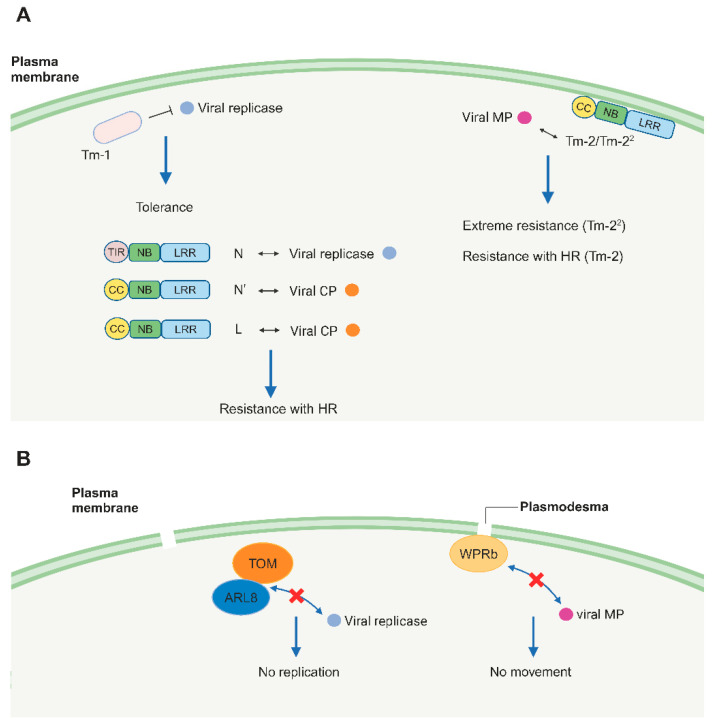
Dominant (**A**) and recessive resistance (**B**) against tobamoviruses. (**A**). In nucleotide-binding leucine-rich repeat receptor (NLR)-mediated resistance, NLR proteins interact with avirulence (Avr) proteins, leading to the hypersensitive response (HR) or extreme resistance. In *Tm-1*-mediated tolerance, Tm-1 interacts with viral replicase to inhibit viral replication. (**B**). Recessive resistance is caused by the absence of a host factor that is essential for tobamovirus infection. Both TOM and ARL8 interact with viral replicase to promote tobamoviruses replication. WPRb interacts with the viral movement protein (MP) to facilitate viral movement. TIR: Toll-interleukin-1 receptor domain, CC: coiled-coil domain, NB: nucleotide-binding domain, LRR: leucine-rich repeat domain, CP: coat protein, TOM: Tobamovirus multiplication, ARL8: Arabidopsis ADP-ribosylation factor-like 8, WPRb: WEB1/PMI2-related protein. Figure adapted from images created with BioRender.com.

**Figure 2 viruses-16-00530-f002:**
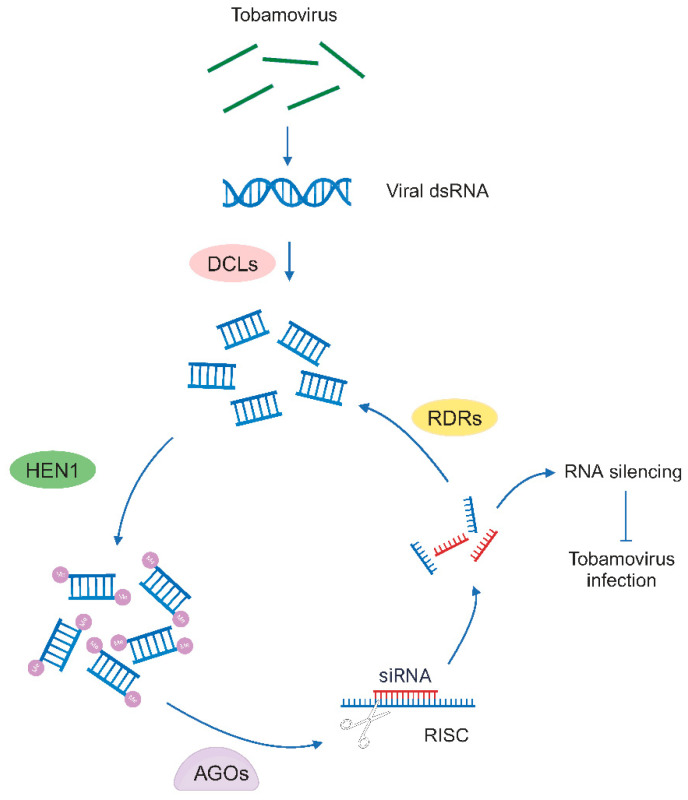
The RNA silencing pathway against tobamoviruses. Upon tobamovirus infection, viral double-stranded RNA (dsRNA) is generated by viral replicase as an intermediate in genome replication, and it is cleaved by Dicer-like (DCL) enzymes to generate viral small RNAs (sRNAs) duplexes of 21–24 nucleotides. The 3′ terminal nucleotides of the sRNAs duplexes are methylated by HUA ENHANCER1 (HEN1) to protect them from degradation. The resulting sRNAs are incorporated into the Argonaute (AGO)-containing RNA-induced silencing complex (RISC) for silencing, thus inhibit the tobamovirus infection. This process is further enhanced by RNA-dependent RNA polymerases (RDRs), which bind to the product of RISC and produce the secondary sRNAs. Figure adapted from images created with BioRender.com.

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
