# Peer review of "Plant Immunity against Tobamoviruses"

_viruses, 2024, doi:10.3390/v16040530_

Round 1

Reviewer 1 Report

Comments and Suggestions for Authors

The review by Zheng and Liu focuses on plant immunity against tobamovirus infection. Given that this work was submitted to a special issue of the journal "Tobamoviruses: Molecular Aspects and Resistance Regulation—a Special Issue Commemorating 125 Years of Research on Tobamoviruses," the theme of plant defense mechanisms against viruses related to the Tobamovirus genus is highly relevant. Undoubtedly, the review is well-written and clear. Despite the recent release of a number of similar reviews on the topic of plant immune defense against plant viruses, with the most recent by Spiegelman and Dinesh-Kumar (2023) (https://www.annualreviews.org/doi/pdf/10.1146/annurev-virology-111821-122847), the authors mentioned at least 10 studies published in the last year (2023-2024), which is undoubtedly a positive aspect of this review. However, I have a few recommendations that could help improve the quality of the article.

1. Unlike the aforementioned review, there is a complete absence of visual representation of the discussed data here. I recommend present some summarized data in the form of diagrams or figures, such as the general known immune mechanisms of plant cell defense against tobamoviruses.

2. In addition to research plant immunity against ONLY tobamoviruses, there are numerous studies and reviews focused on plant immune defense in general, where mechanisms of the plant immune response to VARIOUS viral infections are described. Could the authors include examples of immune response mechanisms against another plant viruses (possibly helical or icosahedral, single-stranded or double-stranded RNA) in the Discussion section that have not been reported or identified, or unrelated for tobamoviruses yet? For example, proteins from the lectin family that confer resistance to a range of plant viruses, as described in Ranf et al., 2015 (https://doi.org/10.1038/ni.3124).

3. Shouldn't the review text include the findings of the study by Zhu et al., 2023 (https://doi.org/10.1093/jxb/erad202)?

Reviewer 2 Report

Comments and Suggestions for Authors

Manuscript viruses-2919416 briefly reviews the latest on plant immunity against tobamoviruses. The article is comprehensive and well written for the most part. It could be strengthened by the addition of a short paragraph on conclusions before the paragraph on future directions. Such a paragraph on conclusions would have the merit to highlight and critical assess the major lessons learned so far before listing new research perspectives.

Minor specific comments:

Line 19: ... family. The Tobamovirus genus ...

Line 21: ... rod-shaped particles ...

Line 25: ... encode a 54 KDa ...

Lines 59-62: A sound explanation should be provided for the inclusion of this sentence because it pertains to a potyvirus. May be suggesting a connection with tobamoviruses would be of interest. Alternatively, the sentence could be eliminated.

Table 1: TNL, CNL and CC should be spelled out.

Line 78: Why is resistance italicized and with a bigger font size?

Line 86: Change recognizing to recognizes

Lines 87-88: ... immune response in the cytoplasm ...

Line 88: ... resistance to TMV in the cell ...

Line 93: ... in the nucleus and detected only ...

Line 155: ... resistance is compomised at ...

Line 155: ... we showed that the N-terminus of ...

Line 157: ... recognizes the N-terminal but not the C-terminal sequence of the viral MP ...

Line 158: ... and the C-terminal domain ...

Lines 248 and 249: Why are exosome and exoribonuclease underlined and written in a bigger font size?

Lines 249-250: Add a space between 3'-5' and direction

Lines 324-325: ... has been restricted for field use. However, it still holds great potential for generating virus resistant plants.

Comments on the Quality of English Language

See minor suggestions for language improvement
